# Central versus Peripheral Postcardiotomy Veno-Arterial Extracorporeal Membrane Oxygenation: Systematic Review and Individual Patient Data Meta-Analysis

**DOI:** 10.3390/jcm11247406

**Published:** 2022-12-14

**Authors:** Fausto Biancari, Alexander Kaserer, Andrea Perrotti, Vito G. Ruggieri, Sung-Min Cho, Jin Kook Kang, Magnus Dalén, Henryk Welp, Kristján Jónsson, Sigurdur Ragnarsson, Francisco J. Hernández Pérez, Giuseppe Gatti, Khalid Alkhamees, Antonio Loforte, Andrea Lechiancole, Stefano Rosato, Cristiano Spadaccio, Matteo Pettinari, Antonio Fiore, Timo Mäkikallio, Sebastian D. Sahli, Camilla L’Acqua, Amr A. Arafat, Monirah A. Albabtain, Mohammed M. AlBarak, Mohamed Laimoud, Ilija Djordjevic, Ihor Krasivskyi, Robertas Samalavicius, Lina Puodziukaite, Marta Alonso-Fernandez-Gatta, Markus J. Wilhelm, Giovanni Mariscalco

**Affiliations:** 1Heart and Lung Center, Helsinki University Hospital, Haartmaninkatu 4, P.O. Box 340, 00029 Helsinki, Finland; 2Department of Medicine, South-Karelia Central Hospital, University of Helsinki, 53130 Lappeenranta, Finland; 3Institute of Anesthesiology, University and University Hospital Zurich, 8091 Zurich, Switzerland; 4Department of Thoracic and Cardio-Vascular Surgery, University Hospital Jean Minjoz, 25030 Besançon, France; 5Division of Cardiothoracic and Vascular Surgery, Robert Debré University Hospital, 51100 Reims, France; 6Divisions of Neurosciences, Critical Care and Cardiac Surgery, Departments of Neurology, Anesthesiology and Critical Care Medicine, Johns Hopkins University School of Medicine, Baltimore, MD 21201, USA; 7Department of Molecular Medicine and Surgery, Department of Cardiac Surgery, Karolinska Institutet, Karolinska University Hospital, 17176 Stockholm, Sweden; 8Department of Cardiothoracic Surgery, Münster University Hospital, 48149 Münster, Germany; 9Department of Cardiac Surgery, Sahlgrenska University Hospital, 41345 Gothenburg, Sweden; 10Department of Cardiothoracic Surgery, University of Lund, 10392 Lund, Sweden; 11Puerta de Hierro University Hospital, 29222 Madrid, Spain; 12Division of Cardiac Surgery, Cardio-Thoracic and Vascular Department, University Hospital of Trieste, 34128 Trieste, Italy; 13Prince Sultan Cardiac Center, Al Hassa 36441, Saudi Arabia; 14Department of Cardiothoracic, Transplantation and Vascular Surgery, S. Orsola Hospital, University of Bologna, 40138 Bologna, Italy; 15Cardiothoracic Department, University Hospital of Udine, 33100 Udine, Italy; 16Center for Global Health, Italian National Institute, 00161 Rome, Italy; 17Department of Cardiovascular Surgery, Mayo Clinic, Rochester, MN 13400, USA; 18Department of Cardiovascular Surgery, Ziekenhuis Oost-Limburg, 3600 Genk, Belgium; 19Department of Cardiac Surgery, Hôpitaux Universitaires Henri Mondor, Assistance Publique-Hôpitaux de Paris, 94000 Creteil, France; 20Anesthesia and Intensive Care Unit, Centro Cardiologico Monzino, 20138 Milan, Italy; 21Anesthesia and Intensive Care Unit, Fondazione IRCCS Istituto Nazionale dei Tumori, 20133 Milan, Italy; 22Adult Cardiac Surgery, Prince Sultan Cardiac Center, Riyadh 12611, Saudi Arabia; 23Cardiothoracic Surgery Department, Tanta University, Tanta 31527, Egypt; 24Cardiology Clinical Pharmacy, Prince Sultan Cardiac Center, Riyadh 12611, Saudi Arabia; 25Intensive Care Department, Prince Sultan Cardiac Center, Riyadh 12611, Saudi Arabia; 26Cardiac Surgical Intensive Care Department, King Faisal Specialist Hospital and Research Center, Riyadh 11564, Saudi Arabia; 27Critical Care Medicine Department, Cairo University, Cairo 12613, Egypt; 28Department of Cardiothoracic Surgery, University Hospital Cologne, 50937 Cologne, Germany; 292nd Department of Anesthesia, Vilnius University Hospital Santaros Klinikos, 08410 Vilnius, Lithuania; 30Clinic of Emergency Medicine, Medical Faculty, Vilnius University, 03101 Vilnius, Lithuania; 31Cardiology Department, University Hospital of Salamanca, Instituto de Investigación Biomédica de Salamanca, 37007 Salamanca, Spain; 32CIBER-CV Instituto de Salud Carlos III, 28029 Madrid, Spain; 33Clinic for Cardiac Surgery, University Heart Center, University and University Hospital Zurich, 8091 Zurich, Switzerland; 34Department of Intensive Care Medicine and Cardiac Surgery, Glenfield Hospital, University Hospitals of Leicester, Leicester LE2 9QP, UK

**Keywords:** extracorporeal membrane oxygenation, ECMO, postcardiotomy, cardiac surgery, central, peripheral

## Abstract

Background: It is unclear whether peripheral arterial cannulation is superior to central arterial cannulation for postcardiotomy veno-arterial extracorporeal membrane oxygenation (VA-ECMO). Methods: A systematic review was conducted using PubMed, Scopus, and Google Scholar to identify studies on postcardiotomy VA-ECMO for the present individual patient data (IPD) meta-analysis. Analysis was performed according to the intention-to-treat principle. Results: The investigators of 10 studies agreed to participate in the present IPD meta-analysis. Overall, 1269 patients were included in the analysis. Crude rates of in-hospital mortality after central versus peripheral arterial cannulation for VA-ECMO were 70.7% vs. 63.7%, respectively (adjusted OR 1.38, 95% CI 1.08–1.75). Propensity score matching yielded 538 pairs of patients with balanced baseline characteristics and operative variables. Among these matched cohorts, central arterial cannulation VA-ECMO was associated with significantly higher in-hospital mortality compared to peripheral arterial cannulation VA-ECMO (64.5% vs. 70.8%, *p* = 0.027). These findings were confirmed by aggregate data meta-analysis, which showed that central arterial cannulation was associated with an increased risk of in-hospital mortality compared to peripheral arterial cannulation (OR 1.35, 95% CI 1.04–1.76, I^2^ 21%). Conclusions: Among patients requiring postcardiotomy VA-ECMO, central arterial cannulation was associated with an increased risk of in-hospital mortality compared to peripheral arterial cannulation. This increased risk is of limited magnitude, and further studies are needed to confirm the present findings and to identify the mechanisms underlying the potential beneficial effects of peripheral VA-ECMO.

## 1. Introduction

Veno-arterial extracorporeal membrane oxygenation (VA-EMO) is an effective salvage therapy for postcardiotomy cardiogenic shock unresponsive to aggressive inotropic treatment [1,2]. However, a large pooled analysis showed that two-thirds of patients treated with postcardiotomy VA-ECMO do not survive to hospital discharge [3]. Appropriate patient selection is a key issue in optimizing the burden of resources associated with this therapy and improving the results [2]. Several treatment strategies, such as access sites for arterial cannulation, left ventricular venting, and preventative strategies of bleeding and thrombosis, may have an impact on the recovery of myocardial function and prevention of end-organ injury in these patients. However, the lack of data from randomized studies on these strategies does not allow an understanding of their potential benefits and harms in this critical setting. The optimal access site for arterial cannulation for VA-ECMO is one of the most controversial topics [4]. A meta-analysis by Mariscalco et al. [5] showed that peripheral cannulation for postcardiotomy VA-ECMO was associated with decreased in-hospital mortality compared to central arterial cannulation. Another meta-analysis, including non-postcardiotomy patients, did not confirm this finding [6]. Because of the limitations of previous aggregate data meta-analyses, this controversial issue was evaluated in the present individual patient data (IPD) meta-analysis. 

## 2. Methods

This study is registered in the PROSPERO registry (CRD42022359392). A literature search was performed in August 2022 through PubMed, Scopus, and Google Scholar. This study was accomplished following the Preferred Reporting Items for Systematic Reviews and Meta-Analyses (PRISMA) guidelines [7] (Appendix A).

To enter this analysis, studies had to fulfill all these inclusion criteria: (1) provide data on patients who required VA-ECMO, only as veno-arterial configuration, after cardiac surgery procedure of any kind, including heart transplantation; (2) include patients aged 18 years or older; (3) be a prospective or retrospective observational investigation; (4) be published in the English language as a full article; (5) include at least 10 patients; (6) include data on pre-VA-ECMO arterial lactate; and (7) be published since 2015.

Articles were ineligible for study inclusion if they (1) did not provide specific information on the type of ECMO used in the study population and its related outcome; (2) did not provide specific information on the timing and site of cannulation of VA-ECMO; (3) did not provide data on arterial lactate at VA-ECMO cannulation; (4) included pediatric patients; (5) reported the use of ECMO configuration other than VA-ECMO therapy for cardiogenic shock. We decided not to include pediatric patients because of the expected marked differences between the pediatric and adult cardiac surgery populations regarding patients’ characteristics and heart diseases. The decision to include only studies reporting on arterial lactate at the time of implantation of VA-ECMO was based on the significant prognostic impact of this biomarker of systemic tissue hypoxia on the early outcome of these critically ill patients [2,3,8,9,10]. The population, intervention, comparison, and outcomes of the present study are summarized in Table 1.

Studies were independently screened by two investigators (F.B., G.M.) through PubMed, Scopus, and Google Scholar using the terms “Postcardiotomy” and “ECMO” or “ECLS”. One or more authors of articles suitable for inclusion in the present IPD meta-analysis were contacted three times by email, and they were provided with a study protocol with the definition criteria of variables of interest and an Excel datasheet with prespecified covariates. Once the investigators provided their dataset, this was checked for completeness and congruency. Patients from these studies who did not fulfill the inclusion criteria were excluded from the analysis. The quality of the included studies was assessed according to the National Heart, Lung, and Blood Institute Study Quality Assessment Tools for case series studies [11]. 

For the purpose of this study, central arterial cannulation for VA-ECMO was considered as an arterial cannulation of the ascending aorta. Peripheral arterial cannulation for VA-ECMO was defined as the cannulation of any peripheral artery. In this analysis, central and peripheral arterial cannulation were defined according to the intention-to-treat principle.

The outcome measure of this study was in-hospital mortality, i.e., all-cause mortality during the index hospitalization.

### Statistical Analysis

Categorical variables were reported as counts and percentages. Continuous variables were reported as means and standard deviations. Risk estimates were reported as odds ratios (OR) with a 95% confidence interval (CI). Univariable analysis of continuous variables was performed using the Mann–Whitney test and of categorical variables using the chi-square test or Fischer’s exact test. Logistic regression using the stepwise backward method was performed with in-hospital mortality as the dependent variables, including the baseline and operative covariates with a *p* < 0.2 in univariable analysis into the regression model (Table 2). Calibration of the regression model was assessed by estimation of the area under the receiver operating characteristics curve (ROC) and discrimination with the Hosmer–Lemeshow’s test. Considering the expected imbalance in the baseline and operative covariates between central and peripheral arterial cannulation study groups, a propensity score matching analysis was performed employing a caliper width of 0.5. The propensity score was calculated with logistic regression considering all the covariates listed in Table 2, except the duration of VA-ECMO therapy. A standardized difference <0.10 was considered an acceptable balance between the covariates of the study groups. The effect of the arterial cannulation site was also estimated using aggregate data meta-analysis with the fixed-effects and random-effects method and the Mantel–Haenszel test. The I^2^ test was used to estimate studies heterogeneity. Statistical analyses were performed with SPSS (version 27.0, SPSS Inc., IBM, Chicago, IL, USA), Stata (version 15.1, StataCorp LLC, College Station, TX, USA), and RevMan 5.4.1 (the Cochrane Collaboration, 2020) statistical software.

## 3. Results

A systematic review of the literature yielded 273 articles. Thirty-one studies were considered suitable for this analysis. The investigators of 10 studies [2,12,13,14,15,16,17,18,19,20] agreed to participate in the present IPD meta-analysis and provided their complete and anonymized data in an Excel datasheet with prespecified variables. The number of patients, type of studies, and quality of the included studies are summarized in Table 2. These studies included 1503 patients, and 234 of them were excluded according to the study exclusion criteria (Figure 1). Overall, 1269 patients (556 with central VA-ECMO and 713 patients with peripheral VA-ECMO) were included in this analysis. Baseline characteristics and operative data of patients who died during the index hospitalization or survived to discharge are summarized in Table 3.

Crude rates of in-hospital mortality after central versus peripheral arterial cannulation for VA-ECMO were 70.7% vs. 63.7%, respectively (*p* = 0.01) (Table 3). In-hospital mortality of patients in whom central arterial cannulation was switched to peripheral arterial cannulation VA-ECMO was 71.9% (23 out of 32 patients), and of those switched from peripheral arterial cannulation to central arterial cannulation VA-ECMO was 62.5% (5 out of 8 patients) (*p* = 0.07). 

Predictors of in-hospital mortality in the univariable analysis are summarized in Table 3. Logistic regression showed that central arterial cannulation for VA-ECMO was associated with a significantly increased risk of in-hospital mortality compared to peripheral cannulation (adjusted OR 1.31, 95% CI 1.02–1.70) (Hosmer–Lemeshow test, *p* = 0.58, area under the ROC curve 0.71, 95% CI 0.68–0.74) (Table 3). 

Propensity score matching yielded 538 pairs of patients with balanced baseline characteristics and operative variables (Table 4). Among these matched cohorts, central arterial cannulation VA-ECMO was associated with significantly higher in-hospital mortality compared to peripheral arterial cannulation VA-ECMO (70.8% vs. 64.5%, *p* = 0.03).

Aggregate data meta-analysis with the fixed-effects method showed that, with low heterogeneity between studies (I^2^ 21% and see funnel plot in Figure 2), central cannulation for postcardiotomy VA-ECMO was associated with increased risk of in-hospital mortality (OR 1.35, 95% CI 1.04–1.76). However, this effect was not significant when the analysis was performed with the random-effects method (OR 1.34, 95% CI 0.93–1.93).

## 4. Discussion

The present findings suggest that the use of peripheral cannulation for postcardiotomy VA-ECMO may reduce the risk of in-hospital mortality in these critically ill patients. These results are based on an intention-to-treat principle, and it is worth noting that only in 8 (1.6%) patients the VA-ECMO configuration was switched from peripheral to central arterial cannulation. However, we do not have data on whether any other VA-ECMO configuration was adopted to avoid upper body hypoxia, the so-called North-South syndrome, or Harlequin syndrome [21]. 

These results are comparable to those of the aggregate data meta-analysis by Mariscalco et al. [5], but they were not consonant with those of another aggregate data meta-analysis by Raffa et al. [6]. However, the latter study might have been biased by including non-postcardiotomy cardiogenic shock in the analysis, which most likely was included in the peripheral cannulation cohort. Furthermore, the present IPD meta-analysis showed that previous studies included several patients who did not fulfill the criteria of postcardiotomy VA-ECMO, and their exclusion from aggregate data meta-analysis is not feasible. Furthermore, IPD meta-analyses, such as the present one, can be adjusted for risk factors that have a significant impact on in-hospital mortality. Indeed, we observed that in addition to the site of arterial cannulation, age, gender, aortic arch surgery, prior cardiac surgery, preoperative acute neurological event, and arterial lactate at VA-ECMO cannulation were independent predictors of poor outcome. Failure to adjust for these risk factors might, therefore, introduce bias.

One recent study by Radakovic et al. [22] showed that central arterial cannulation was associated with improved survival compared to peripheral cannulation. A series of small sample size by Merritt–Genore et al. [23] confirmed that central arterial cannulation was associated with improved survival after postcardiotomy VA-ECMO. However, neither study was adjusted for potentially relevant risk factors, and they were not adequately powered.

A study by Kalampokas [24] showed improved results with the peripheral arterial cannulation for postcardiotomy VA-ECMO compared to central cannulation, but the benefit of the peripheral approach vanished in 20 propensity score-matched pairs, likely because of the small sample size. Indeed, the main problem is the lack of data from adequately powered studies. In fact, analysis from a large sample size population from the Extracorporeal Life Support Organization (ELSO) demonstrated that peripheral cannulation was associated with 50% risk reduction compared to central cannulation for postcardiotomy VA-ECMO (OR 0.48; 95% CI 0.40–0.58) [1].

The present study was not planned to identify the mechanisms underlying the potential benefits of the vascular access site for VA-ECMO. However, we hypothesize that peripheral cannulation may be beneficial by reducing the risk of severe intrathoracic bleeding [25]. Indeed, in the meta-analysis by Mariscalco et al. [5], peripheral VA-ECMO significantly reduced the risk of reoperation for bleeding/tamponade (risk ratio, 0.63; 95% CI, 0.54–0.73), while the meta-analysis by Raffa et al. [6] demonstrated a significant reduction in the risk of reoperation for bleeding (OR 0.65, 95% CI 0.46–0.93). Importantly, the authors showed that peripheral VA-ECMO was associated with a reduction of seven units of red blood cell transfusion [6]. Excessive bleeding requiring re-exploration and the use of a large amount of blood transfusion has been shown to have a significant impact on early and late mortality after adult cardiac surgery [26] and may explain why peripheral VA-ECMO may significantly reduce the risk of in-hospital mortality. Raffa et al. [6] also demonstrated that peripheral VA-ECMO also contributed to a significant reduction in terms of continuous venovenous hemofiltration (OR 0.76, 95% CI 0.60–0.97), which may further reduce the risk of early mortality [27]. 

Several limitations may affect the results of this study. First, all included data were from retrospective cases series. Second, the results might be biased by heterogeneity in the experience and treatment strategies of VA-ECMO patients. Indeed, it is difficult to disentangle the effect of hospital expertise and the site of cannulation. Third, only a limited number of studies contributed to this IPD meta-analysis. Fourth, aggregate data meta-analysis confirmed the present findings only when the fixed-effects method was used, but only a trend toward a beneficial effect of peripheral VA-ECMO was noted with the random-effects method. However, we did not detect significant heterogeneity in the I^2^ test, and this was confirmed with the funnel plot. This lack of heterogeneity justifies the use of the fixed-effects method in this setting, which likely gave higher weight to larger studies. Finally, our study selection was based on the need for data granularity. In particular, we decided to consider for this IPD meta-analysis only studies reporting on pre-VA-ECMO arterial lactate. This decision was based on the recent knowledge of the prognostic significance of arterial lactate as a marker of systemic tissue hypoxia in these critically ill patients [2,3,8,9,10]. 

The strength of this study resides in the rather large size of this series. In fact, post hoc power analysis for matched series showed that 285 patients per study group would have been enough to reject the null hypothesis (alpha 0.05, beta 0.80). Furthermore, the availability of data on variables of relevance, such as pre-VA-ECMO lactate and operative data, provided further strength to this analysis. 

In conclusion, the results of this IPD meta-analysis showed that, among patients requiring postcardiotomy VA-ECMO, central arterial cannulation was associated with a significantly increased risk of in-hospital mortality compared to peripheral arterial cannulation. Such a risk is of limited magnitude, and further studies are needed to confirm the present findings and to identify the reasons for the improved outcome of postcardiotomy VA-ECMO with the peripheral arterial approach.

## Figures and Tables

**Figure 1 jcm-11-07406-f001:**
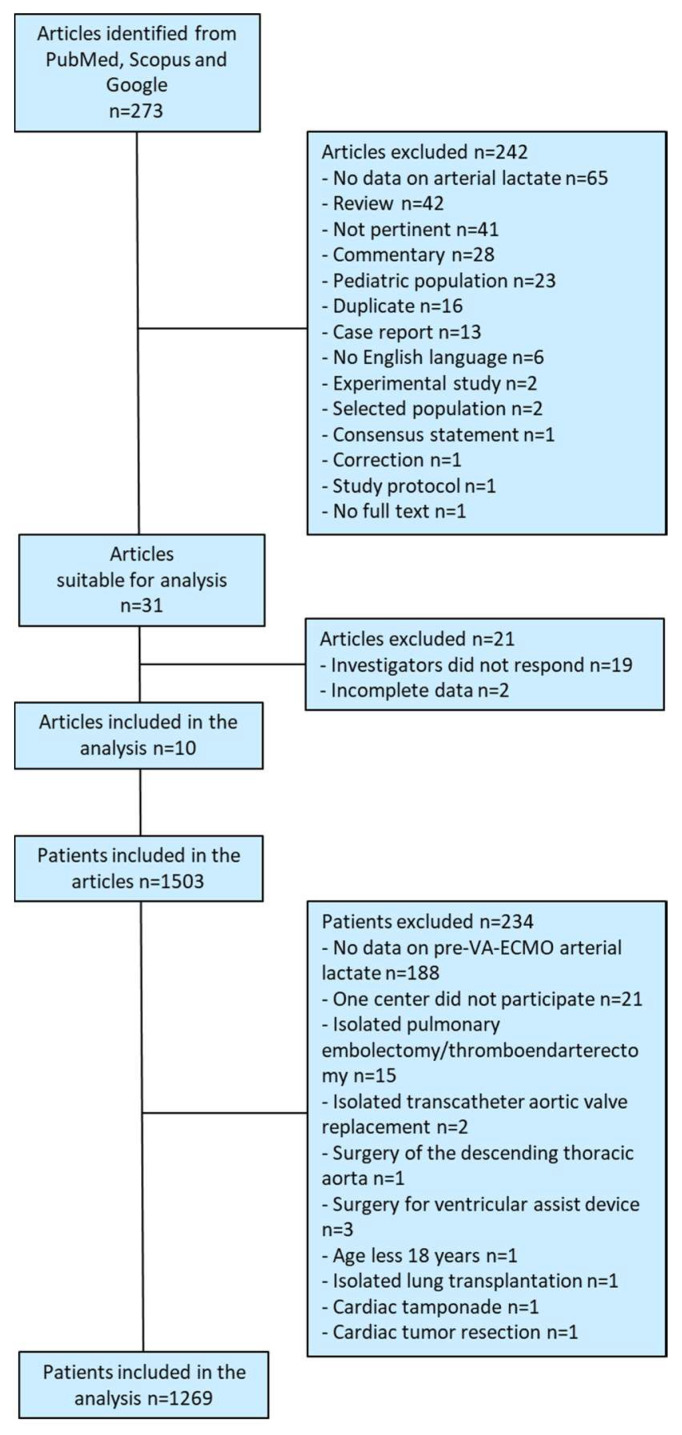
Study flow chart.

**Figure 2 jcm-11-07406-f002:**
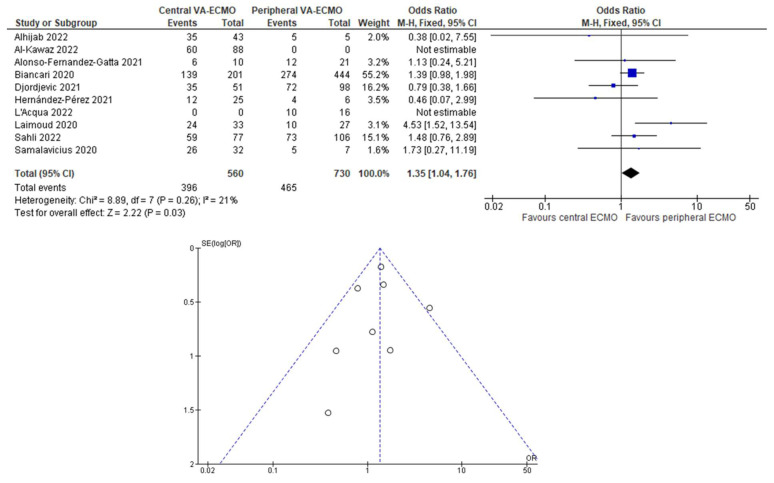
Forest plot and funnel plot from aggregate data meta-analysis [2,12,13,14,15,16,17,18,19,20].

**Table 1 jcm-11-07406-t001:** Population, intervention, comparison, and outcomes of the present study.

**Population**	Patients who underwent any adult cardiac surgery procedure
**Intervention**	Veno-arterial extracorporeal membrane oxygenation for cardiogenic shock after adult cardiac surgery
**Comparison**	Central versus peripheral arterial cannulation for veno-arterial extracorporeal membrane oxygenation
**Outcomes**	Mortality during the index hospitalization, mortality during veno-arterial extracorporeal membrane oxygenation

**Table 2 jcm-11-07406-t002:** Characteristics and quality of studies included in the present individual patient data meta-analysis according to the National Heart, Lung, and Blood Institute Study Quality Assessment Tools for case series studies.

	Al-Kawaz 2022 [12]	Alhijab 2022 [13]	Alonso-Fernandez-Gatta 2021 [14]	Biancari 2020 [2]	Djordjevic 2021 [15]	Hernández-Pérez 2021 [16]	L’Acqua 2022 [17]	Laimoud 2020 [18]	Sahli 2022 [19]	Samalavicius 2020 [20]
No. of patients *	50	101	34	781	172	32	17	61	215	40
No. of patients included in the analysis	48	88	31	645	149	31	16	60	183	39
Multicenter study	No	No	No	Yes	No	No	No	No	No	No
Prospective study	No	No	No	No	No	No	No	No	No	No
NHLBI study quality criteria										
1. Was the study question or objective clearly stated?	Yes	Yes	Yes	Yes	Yes	Yes	Yes	Yes	Yes	Yes
2. Was the study population clearly and fully described, including a case definition?	Yes	Yes	Yes	Yes	Yes	Yes	No	Yes	Yes	Yes
3. Were the cases consecutive?	No	Yes	Yes	Yes	Yes	Yes	Yes	Yes	Yes	Yes
4. Were the subjects comparable?	Yes	Yes	Yes	Yes	Yes	Yes	Yes	Yes	Yes	No
5. Was the intervention clearly described?	Yes	Yes	Yes	Yes	Yes	Yes	Yes	Yes	Yes	Yes
6. Were the outcome measures clearly defined, valid, reliable, and implemented consistently across all study participants?	Yes	Yes	Yes	Yes	No	Yes	No	Yes	Yes	No
7. Was the length of follow-up adequate?	Yes	Yes	Yes	Yes	Yes	Yes	Yes	Yes	Yes	Yes
8. Were the statistical methods well-described?	Yes	Yes	Yes	Yes	Yes	Yes	Yes	Yes	Yes	Yes
9. Were the results well-described?	Yes	Yes	Yes	Yes	Yes	Yes	No	Yes	Yes	Yes
Quality rating	Good	Good	Good	Good	Good	Good	Fair	Good	Good	Fair

* The overall number of patients requiring postcardiotomy veno-arterial extracorporeal membrane oxygenation as included in the original article. NHLB = National Heart, Lung, and Blood Institute.

**Table 3 jcm-11-07406-t003:** Patients’ characteristics, operative data, and their impact on in-hospital mortality.

Clinical Variables	Alive N = 422	In-Hospital Death N = 847	Univariable Analysis *p*-Value	Multivariate Analysis
Baseline characteristics				
Age, years	58.2 (14.5)	63.8 (13.2)	<0.0001	1.03, 1.02–1.04
Female gender	109 (25.8)	290 (34.2)	0.002	1.43, 1.09–1.88
eGFR, mL/min/1.73 m^2^	74 (34)	65 (32)	<0.0001	
Coronary artery disease	203 (48.1)	392 (46.3)	0.540	
Type A aortic dissection	23 (5.5)	73 (8.6)	0.044	
Preop. acute neurological event	14 (3.3)	60 (7.1)	0.007	2.13, 1.14–3.97
Prior cardiac surgery	83 (19.7)	224 (26.4)	0.008	1.55, 1.14–2.10
Arterial lactate, mmol/L	6.7 (4.3)	9.3 (5.9)	<0.0001	1.11, 1.08–1.14
Procedural data				
Urgent/emergency surgery	206 (48.8)	430 (50.8)	0.512	
Isolated CABG	98 (23.2)	188 (22.2)	0.680	
Any CABG	200 (47.4)	397 (46.9)	0.861	
Aortic valve procedure	164 (38.9)	296 (34.9)	0.172	
Mitral valve procedure	151 (35.8)	297 (35.1)	0.801	
Tricuspid valve procedure	54 (12.8)	125 (14.8)	0.344	
Pulmonary valve procedure	2 (0.5)	4 (0.5)	1.000	
VSD or ventricular wall repair	14 (3.3)	30 (3.5)	0.837	
Septal myectomy	1 (0.2)	6 (0.7)	0.285	
Aortic surgery	73 (17.3)	172 (20.3)	0.201	
Aortic root replacement	44 (10.4)	84 (9.9)	0.777	
Aortic arch surgery	9 (2.1)	54 (6.4)	0.001	2.94, 1.40–6.20
Heart/heart and lung transplantation	8 (1.9)	23 (2.7)	0.373	
Other procedures	15 (3.6)	53 (6.3)	0.044	
VA-ECMO at primary surgery	268 (63.7)	506 (60.0)	0.203	
IABP during VA-ECMO	172 (40.8)	347 (41.0)	0.930	
Central VA-ECMO	163 (38.6)	393 (46.4)	0.009	1.31, 1.02–1.70
VA-ECMO duration, days	7.0 (5.7)	6.2 (6.8)	<0.0001	

Continuous variables are means and standard deviations. Categorical variables are counts and percentages. CABG = coronary artery bypass grafting; eGFR = estimated glomerular filtration rate; IABP = intra-aortic balloon pump; VA-ECMO = veno-arterial extracorporeal membrane oxygenation.

**Table 4 jcm-11-07406-t004:** Patients’ characteristics and operative data in unmatched and propensity score matched pairs.

	Unmatched Patients			Propensity Score Matched Patients		
Clinical Variables	Peripheral ECMO N = 713	Central ECMO N = 556	Standardized Difference	Peripheral ECMO N = 538	Central ECMO N = 538	Standardized Difference
Baseline characteristics						
Age, years	62.6 (13.0)	60.9 (15.0)	0.118	62.3 (13.1)	61.1 (14.8)	0.083
Septuagenarian	224 (31.4)	178 (32.0)	0.013	167 (31.0)	173 (32.2)	0.024
Female gender	207 (29.0)	192 (34.5)	0.118	171 (31.8)	183 (34.0)	0.047
eGFR, mL/min/1.73 m^2^	67 (31)	69 (36)	0.063	68 (32)	69 (36)	0.029
Coronary artery disease	334 (46.8)	261 (46.9)	0.002	253 (47.0)	254 (47.2)	0.004
Type A aortic dissection	54 (7.6)	42 (7.6)	0.001	44 (8.2)	42 (7.8)	0.013
Preop. acute neurological event	48 (6.7)	26 (4.7)	0.089	28 (5.2)	26 (4.8)	0.017
Prior cardiac surgery	169 (23.7)	138 (24.8)	0.026	133 (24.7)	130 (24.2)	0.013
Arterial lactate, mmol/L	7.9 (5.2)	9.2 (5.8)	0.233	8.6 (5.3)	9.0 (5.6)	0.076
Procedural data						
Urgent/emergency surgery	386 (54.1)	250 (45.0)	0.182	262 (48.7)	243 (45.2)	0.071
Isolated CABG	161 (22.6)	125 (22.5)	0.002	130 (24.2)	123 (22.9)	0.031
Any CABG	328 (46.0)	269 (48.4)	0.048	259 (48.1)	261 (48.5)	0.007
Aortic valve procedure	262 (36.7)	198 (35.6)	0.024	195 (36.2)	192 (35.7)	0.012
Mitral valve procedure	238 (33.4)	210 (37.8)	0.092	187 (34.8)	203 (37.7)	0.062
Tricuspid valve procedure	94 (13.2)	85 (15.3)	0.060	73 (13.6)	83 (15.4)	0.053
Pulmonary valve procedure	1 (0.1)	5 (0.9)	0.110	1 (0.2)	2 (0.4)	0.036
VSD or ventricular wall repair	31 (4.3)	13 (2.3)	0.112	15 (2.8)	13 (2.4)	0.023
Septal myectomy	2 (0.3)	5 (0.9)	0.081	1 (0.2)	2 (0.4)	0.036
Aortic surgery	138 (19.4)	107 (19.2)	0.003	107 (19.9)	106 (19.7)	0.005
Aortic root replacement	71 (10.0)	57 (10.3)	0.010	54 (10.0)	57 (10.6)	0.018
Aortic arch surgery	34 (4.8)	29 (5.2)	0.021	30 (5.6)	29 (5.4)	0.008
Heart/heart and lung transplantation	10 (1.4)	21 (3.8)	0.150	10 (1.9)	18 (3.3)	0.094
Other procedures	29 (4.1)	39 (7.0)	0.129	24 (4.5)	37 (6.9)	0.011
VA-ECMO at primary surgery	411 (57.9)	363 (65.4)	0.155	332 (61.7)	349 (64.9)	0.066
IABP during VA-ECMO	258 (3.2)	261 (47.0)	0.221	226 (42.0)	249 (46.3)	0.086
VA-ECMO duration, days	6.6 (6.2)	6.2 (6.6)	0.057	6.7 (6.4)	6.2 (6.5)	0.073

Continuous variables are means and standard deviations. Categorical variables are counts and percentages. CABG = coronary artery bypass grafting; eGFR = estimated glomerular filtration rate; IABP = intra-aortic balloon pump; VA-ECMO = veno-arterial extracorporeal membrane oxygenation; VSD = ventricular septal defect.

## Data Availability

Not available because of privacy issues.

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
