# Peer review of "Central versus Peripheral Postcardiotomy Veno-Arterial Extracorporeal Membrane Oxygenation: Systematic Review and Individual Patient Data Meta-Analysis"

_jcm, 2022, doi:10.3390/jcm11247406_

Round 1
Reviewer 1 Report
Dear authors thank you for very good study. According to PRISMA 2020 for Abstract Checklist in Abstract section there is provided sufficient information about study objectives, methods of inclusion and exclusion criteria for this review. The sources of information has lack of risk of possible bias specification. The results are synthetically presented. There is conclusion as the interpretation of the results. The abstract section is well written and sufficient in my opinion
Introduction
The authors described rationale for the review existing knowledge and provided addressed questions. The databases used for searching eligible studies were specified and inclusion or exclusion criteria were described.
Results are clearly presented, discussion is relevant to the results. The authors fulfil the criteria of PRISMA 2020 checklist for authors.
In my opinion this study is after minor changes. I think that scientific, and practical potential is high.
Author Response
Response: We are grateful to the Reviewer for her/his kind comments. We would like to inform this Reviewer that we made completely new analyses because one investigator provided us with data which were not originally published (this can not be accepted because of the nature of individual patient data meta-analysis). Furthermore, other investigators meantime provided the data on 172 patients who were previously published. The results have slightly changed because the difference between central and peripheral cannulation was mitigated after including one study. We modified the text and Conclusions accordingly proposing that further studies are needed on this topic.
Reviewer 2 Report
The authors performed a per-patient meta-analysis of post-cardiectomy VA-ECMO of ten observational studies totaling 1153 eligible patients according to the predefined inclusion criteria. They found that peripheral cannulation was associated with a reduced risk for in-hospital mortality compared to central cannulation.
These data are much welcomed as, currently, it is unclear which cannulation method should be preferred. Of course, the results must be taken with caution as it is based on relatively small retrospected observational single-center studies, and propensity score matching for critically ill patients is profoundly unreliable. Still, at the moment, this is the best we have.
I have several questions and suggestions that might improve the paper.
Were all procedures performed at the OR (by the cardiac surgeon) or some at a later moment, e.g., at the ICU (by the intensivist)?
Table 3 shows all the baseline characteristics used for the logistic regression and propensity score matching. Were there no missing data for these variables? Well-known predictors for hospital mortality like EURO-score or APACHE-score are lacking. Please clarify.
Do the authors have any information on what may have caused the increased mortality in central cannulated patients? Bleeding and reoperation, as suggested in the Discussion, for example?
Consider restricting decimals for ORs but also p values to two. It will increase readability.
Author Response
Note to the Reviewer: Response: We are grateful to the Reviewer for her/his kind comments. We would like to inform this Reviewer that we made completely new analyses because one investigator provided us with data which were not originally published (this can not be accepted because of the nature of individual patient data meta-analysis). Furthermore, other investigators meantime provided the data on 172 patients who were previously published. The results have slightly changed because the difference between central and peripheral cannulation was mitigated after including one study. We modified the text and Conclusions accordingly proposing that further studies are needed on this topic.
1. Were all procedures performed at the OR (by the cardiac surgeon) or some at a later moment, e.g., at the ICU (by the intensivist)? Table 3 shows all the baseline characteristics used for the logistic regression and propensity score matching. Were there no missing data for these variables? Well-known predictors for hospital mortality like EURO-score or APACHE-score are lacking. Please clarify. Do the authors have any information on what may have caused the increased mortality in central cannulated patients? Bleeding and reoperation, as suggested in the Discussion, for example?
Response: We do agree with the Reviewer on the importance of this information. Unfortunately, the nature of individual patient data meta-analysis does not allow to get granular data, which would have been informative. This Reviewer may certainly understand that gathered data was from previously published data and further data collection was not performed. We should have performed a multicenter study asking for such relevant variables, but this was not possible with the present study design.
Changes: None.
2. Consider restricting decimals for ORs but also p values to two. It will increase readability.
Response: We made these changes as kindly suggested.
Changes: We revised the text and we limited the decimals of OR and p-values to two decimals.